# Masked Space-Time Hash Encoding for Efficient Dynamic Scene Reconstruction

**Feng Wang**[*1]    **Zilong Chen**[*1]    **Guokang Wang**[1]    **Yafei Song**[2]    **Huaping Liu**[†1]

[1]Beijing National Research Center for Information Science and Technology(BNRist),
Department of Computer Science and Technology, Tsinghua University
[2]Alibaba Group
`wang-f20@mails.tsinghua.edu.cn, hpliu@tsinghua.edu.cn`

## Abstract

In this paper, we propose the **M**asked **S**pace-**T**ime **H**ash encoding (MSTH), a novel method for efficiently reconstructing dynamic 3D scenes from multi-view or monocular videos. Based on the observation that dynamic scenes often contain substantial static areas that result in redundancy in storage and computations, MSTH represents a dynamic scene as a weighted combination of a 3D hash encoding and a 4D hash encoding. The weights for the two components are represented by a learnable mask which is guided by an uncertainty-based objective to reflect the spatial and temporal importance of each 3D position. With this design, our method can reduce the hash collision rate by avoiding redundant queries and modifications on static areas, making it feasible to represent a large number of space-time voxels by hash tables with small size. Besides, without the requirements to fit the large numbers of temporally redundant features independently, our method is easier to optimize and converge rapidly with only twenty minutes of training for a 300-frame dynamic scene. As a result, MSTH obtains consistently better results than previous methods with only 20 minutes of training time and 130 MB of memory storage. Code is available at `https://github.com/masked-spacetime-hashing/msth`.

## 1 Introduction

Neural radiance fields [57, 96, 79, 58, 14] have achieved great success in reconstructing static 3D scenes. The learned volumetric representations can produce photo-realistic rendering for novel views. However, for dynamic scenes, the advancements lag behind that of static scenes due to the inherent complexities caused by the additional time dimension. Specifically, reconstructing dynamic scenes requires more time, memory footprint, and additional temporal consistency compared to static scenes. These factors have resulted in unsatisfactory rendering qualities. Recently, many works [40, 38, 75, 83, 1, 10, 72] have made great progress in dynamic scene reconstruction, improving the efficiency and effectiveness of reconstructions. However, there is still considerable room

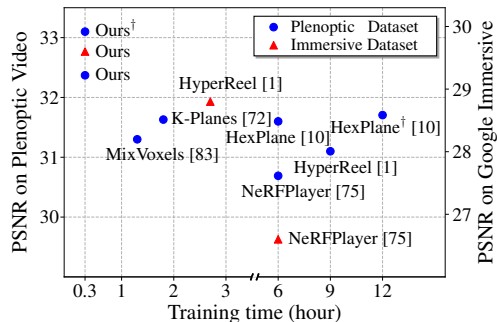

Figure 1: Performance comparison. We compare the PSNR and training time on two public benchmarks. Our method surpasses other methods by a non-trivial margin with only 20m of training. † denotes the HexPlane [10] setting which removes the coffee-martini scene.

---

* Equal contribution.
† Corresponding Author.

37th Conference on Neural Information Processing Systems (NeurIPS 2023).

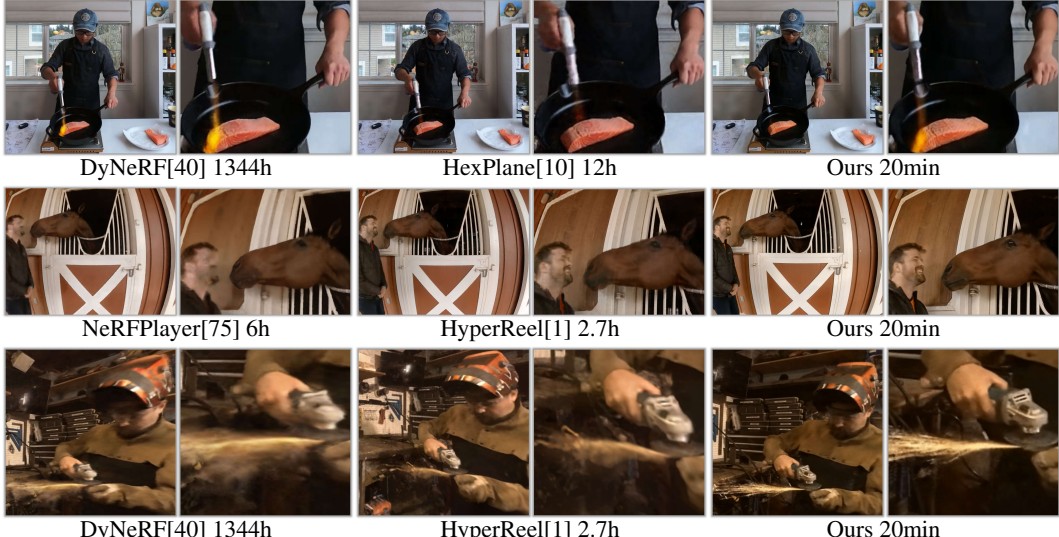

| DyNeRF[40] 1344h | HexPlane[10] 12h | Ours 20min |

| NeRFPlayer[75] 6h | HyperReel[1] 2.7h | Ours 20min |

| DyNeRF[40] 1344h | HyperReel[1] 2.7h | Ours 20min |

Figure 2: Qualitative comparisons to state-of-the-art methods [40, 1, 75, 10]. We visualize three scenes: *flame salmon*, *horse*, and *welder* from Plenoptic Video dataset [40] and Google Immersive dataset [9]. Some key patches are zoomed in for better inspection. Our method performs better in reconstructing details, such as the stripes of the salmon, the facial features, and the splashing sparks.

for improvement in many aspects such as rendering quality, motion coherence, training and rendering speed, and memory usage.

This paper aims to develop a method for reconstructing dynamic scenes with high rendering quality in a space- and time-efficient manner. To achieve this goal, we base our approach on the multi-resolution hash encoding [58] due to its efficiency and compactness for representing and reconstructing static scenes. However, directly applying a 4D multi-resolution hash table to represent a dynamic scene would require a much larger hash table size than that for a static scene due to the much more hash collisions caused by the additional time dimension. Our finding is that, for the ubiquitous existence of static areas in a scene, storing the static points into a 4D time-dependent hash table will result in an information redundancy since each of them will occupy $\mathcal{T}$ hash table items with an identical value for a $\mathcal{T}$-frame scene. It will also lead to a high hash collision rate since it narrows the capacity of the hash table, which negatively impacts the reconstruction quality. Therefore, for the points with low dynamics, we hope to establish a mechanism to reduce the frequent queries and updates to the 4D hash table and automatically save their features into a 3D hash table to avoid temporal redundancy.

From this observation, we propose Masked Space-Time Hash encoding, which combines a 3D hash encoder and a 4D hash encoder with a 3D learnable weight mask. In order to make the mask correlate with the dynamics of the corresponding positions, we adopt the Bayesian framework of Kendall and Gal [34] to estimate the uncertainty [34, 54] of each point being static. The non-linear correlation [6] between the uncertainty and the learnable mask is maximized to make the mask reflect the dynamics. In this way, the static points indicated with a low uncertainty will have a low weight for the 4D hash table and a high weight for the 3D hash table, which prevents modifications to the dynamic representations. With the proposed masked space-time hash encoding, we can set the size of a 4D hash table the same as a 3D one without much loss of rendering qualities, making the representation highly compact. Besides, without the requirements to fit the large numbers of repeated features independently, our method is easier to optimize and converge rapidly in only twenty minutes. To validate the effectiveness of our method on scenes in more realistic settings with large areas of dynamic regions and more complex movements, we collect a synchronized multi-view video dataset with 6 challenging dynamic scenes, which will be publicly available. As a result, the proposed masked space-time hash encoding achieves consistently better reconstruction metrics on two publicly available datasets consisting of 13 scenes, with only *20 minutes* of training and *130 MB* of storage. Fig. 1 and Fig. 2 show the quantitative and qualitative comparisons to other state-of-the-art methods. In summary, our contributions are:

- We propose Masked Space-time Hash Encoding, a novel representation that decomposes the dynamic radiance fields into a weighted combination of 3D and 4D hash encoders.

- The proposed method is validated on a wide range of dynamic scenes, surpassing previous state-of-the-art methods by non-trivial margins with only 20 minutes of training time and 130 MB of memory storage.

- We propose a new synchronized multi-view dataset with more challenging dynamic scenes, including scenes with many moving objects and scenes with complex and rapid movements.

## 2 Related Work

**Neural Scene Representations for Static Scenes.** Representing 3D scenes with neural networks has achieved remarkable success in recent years. NeRF [57] first proposes to use neural networks to represent radiance fields, demonstrating the remarkable potential of this representation in conjunction with volume rendering techniques. The high-fidelity rendering has sparked a surge of interest in related areas. Numerous variants have emerged to address the inherent challenges of the original approach, including improving rendering quality [3, 5, 35, 52, 90], handling sparse input [60, 13, 92, 55, 94, 17], lighting [8, 76, 62], editing [37, 48, 45, 20, 56], dealing with complicated [86, 25, 95, 36] and large scenes [81, 100, 54, 15], generalization [98, 73, 80, 12, 87, 59, 29, 88], jointly optimizing with camera poses [43, 89, 7, 30, 78, 103, 16], accelerating training [96, 14, 79, 93, 18] and speeding up rendering [71, 26, 97, 70, 47, 69, 44, 24, 28, 11, 84, 51]. Our work draws inspiration from two recent contributions in the field, namely Instant-NGP [58] and Mip-NeRF 360 [4]. In Instant-NGP, Müller et al. [58] proposes a novel data structure that leverages multi-resolution hash grids to efficiently encode the underlying voxel grid while addressing hash collisions using a decoding MLP. Furthermore, the proposed method allows for fast and memory-efficient rendering of large-scale scenes. Similarly, Mip-NeRF 360 [4] presents a sample-efficient scheme for unbound scenes, which utilizes a small density field as a sample generator and parameterizes the unbounded scene with spherical contraction. In this paper, we build upon these contributions and propose a novel approach that extends successful techniques and components to incorporate time dimension with minimal overhead. Our method is capable of representing a dynamic scene with only 2~3× memory footprint than that of a static NeRF.

**Novel View Synthesis for Dynamic Scenes.** Extending the neural radiance field to express dynamic scenes is a natural yet challenging task that has been proven crucial to many downstream applications [39, 66, 77, 31]. Many researchers focus on monocular dynamic novel view synthesis, which takes a monocular video as input and targets to reconstruct the underlying 4D information by modeling deformation explicitly [19, 22, 41, 91, 68, 99] or implicitly [64, 23, 63, 21, 74, 49, 46, 102, 82, 32]. As an exemplary work, D-NeRF [68] models a time-varying field through a deformation network that maps a 4D coordinate into a spatial point in canonical space. Despite the great outcomes achieved by research in this line, the applicability of these methods is restricted by the inherent nature of the underlying problem [23]. A more practical way of reconstructing dynamic scenes is by employing multi-view synchronized videos [33, 50, 104, 38, 1, 75, 10, 83, 85, 2, 67]. DyNeRF [40] models dynamic scenes by exploiting a 6D plenoptic MLP with time queries and a set of difference-based importance sampling strategies. The authors also contributed to the field by presenting a real-world dataset, which validated their proposed methodology and provided a valuable resource for future research endeavors. K-Planes [72] and HexPlane [10] speed up training by decomposing the underlying 4D radiance field into several low-dimensional planes, which substantially reduces the required memory footprint and computational complexity compared with explicit 4D voxel grid. HyperReel [1] breaks down the videos into keyframe-based segments and predicts spatial offset towards the nearest keyframe. NeRFPlayer [75] and MixVoxel [83] address the problem by decomposing 4D space according to corresponding temporal properties. The former separates the original space into three kinds of fields and applies different structures and training strategies, while the latter decouples static and dynamic voxels with a variational field which further facilitates high-quality reconstruction and fast rendering. Our method implicitly decomposes 3D space with a learnable mask. This mask eliminates the requirement for manual determination of dynamic pixels and enables the acquisition of uncertainty information, facilitating high-quality reconstruction.

# 3 Methodology

Our objective is to reconstruct dynamic scenes from a collection of multi-view or monocular videos with a compact representation while achieving fast training and rendering speeds. To this end, we propose Masked Space-Time Hash encoding, which represents a dynamic radiance field by a weighted combination of a 3D and a 4D hash encoder. We employ an uncertainty-guided mask learning strategy to enable the mask to learn the dynamics. In the following, we will first introduce the preliminary, then the masked space-time hash encoding and uncertainty-guided mask learning, and finally the ray sampling method.

## 3.1 Preliminary

For neural radiance fields, the input is a 3D space coordinate $\boldsymbol{x}$ and a direction $\boldsymbol{d}$ to enable the radiance field to represent the non-Lambertian effects. The output is the color $c(\boldsymbol{x}, \boldsymbol{d}) \in \mathbb{R}^3$ and a direction-irrelevant density $\sigma(\boldsymbol{x}) \in \mathbb{R}$. Most existing methods encode the input coordinate with a mapping function $h$ that maps the raw coordinate into Fourier features [57], voxels-based features [79, 96], hashing-based features [58] or factorized low-rank features [14]. In this work, we mainly focus on the multi-resolution hash encoding due to its efficiency and compactness. Specifically, for an input point $\boldsymbol{x}$, the corresponding voxel index is computed through the scale of each level, and a hash table index is computed by a bit-wise XOR operation [61]. There are $L$ levels of hash tables corresponding to different grid sizes in a geometric progressive manner. For each level, the feature for a continuous point is tri-linearly interpolated by the nearest grid points. The corresponding features in different levels are concatenated to obtain the final encoding. For convenience, we denote the output of the multi-resolution hash encoder for $\boldsymbol{x}$ as $\mathbf{enc}(\boldsymbol{x})$.

After the multi-resolution hash encoder, a density-decoding MLP $\phi_{\theta_1}$ is employed to obtain the density and an intermediate feature $\boldsymbol{g}$ such that: $\sigma(\boldsymbol{x}), \boldsymbol{g}(\boldsymbol{x}) = \phi_{\theta_1}(\mathbf{enc}(\boldsymbol{x}))$. Then, the color is computed through a color-decoding MLP $\psi_{\theta_2}$: $c(\boldsymbol{x}) = \psi_{\theta_2}(\boldsymbol{g}(\boldsymbol{x}), \boldsymbol{d})$ which is direction-dependent.

For rendering, points along a specific ray $\boldsymbol{r}(s) = \boldsymbol{o} + s\boldsymbol{d}$ are sampled and the volumetric rendering [57] is applied to get the rendered color $\hat{C}(\boldsymbol{r})$:

$$\hat{C}(\boldsymbol{r}) = \int_n^f T(s) \cdot \sigma(\boldsymbol{r}(s)) \cdot c(\boldsymbol{r}(s), \boldsymbol{d}) \mathrm{d}s, \text{ where } T(s) = \exp\left(-\int_{s_n}^s \sigma(\boldsymbol{r}(s)) ds\right). \tag{1}$$

A squared error between the rendered color $\hat{C}(\boldsymbol{r})$ and the ground truth color $C(\boldsymbol{r})$ is applied for back-propagation.

## 3.2 Masked Space-Time Hashing

For a dynamic neural radiance field, the input is a 4D space-time coordinate $(\boldsymbol{x}, t)$. A straightforward method is to replace the 3D hash table with a 4D one. However, this simple replacement will result in a high hash collision rate due to the enormous volume of hash queries and modifications brought by the additional time dimension. For instance, in a dynamic scene comprising $\mathcal{T}$ frames, the hash collision rate is $\mathcal{T}$ times higher than a static scene, leading to a degradation in the reconstruction performance. Enlarging the size of the hash table will cause an unbearable model size and be difficult to scale up with the frame numbers.

To solve this problem, we propose the masked space-time hash encoding, which incorporates a 3D multi-resolution hash mapping $\mathrm{h}_{\mathsf{3D}}$, a 4D multi-resolution hash mapping $\mathrm{h}_{\mathsf{4D}}$, and a learnable mask encoding $\tilde{m}$. The final encoding function is formulated as follows:

$$\mathbf{enc}(\boldsymbol{x}, t) = m(\boldsymbol{x}) \cdot \mathrm{h}_{\mathsf{3D}}(\boldsymbol{x}) + (1 - m(\boldsymbol{x})) \cdot \mathrm{h}_{\mathsf{4D}}(\boldsymbol{x}, t), \text{ where } m(\boldsymbol{x}) = \mathrm{sigmoid}(\tilde{m}(\boldsymbol{x})). \tag{2}$$

The learnable mask $m$ can be represented by a multi-resolution hash table or a 3D voxel grid. For the 4D hash table, the sizes of time steps also adopt a geometric growing multi-resolution scheme, which is reasonable due to the natural hierarchical properties of motions in time scales.

After obtaining the space-time encoding, the density-decoding and color-decoding MLPs are applied to obtain the final outputs:

$$\sigma(\boldsymbol{x}, t), \boldsymbol{g}(\boldsymbol{x}, t) = \phi_{\theta_1}(\mathrm{enc}(\boldsymbol{x}, t)), \quad c(\boldsymbol{x}, \boldsymbol{d}, t) = \psi_{\theta_2}(\boldsymbol{g}(\boldsymbol{x}, t), \boldsymbol{d}). \tag{3}$$

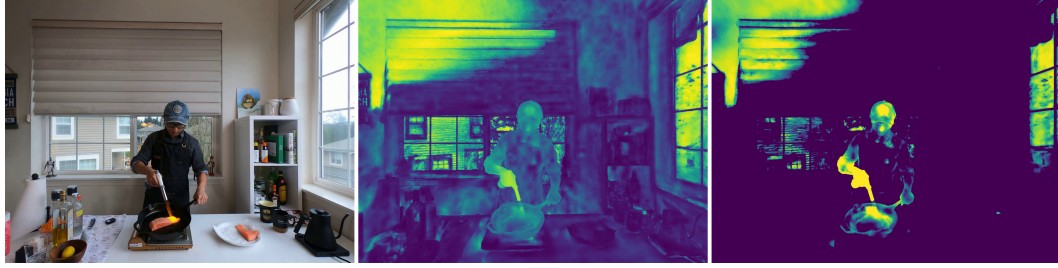

|(a) Origin Image | (b) Mask without Uncertainty | (c) Mask with Uncertainty|

Figure 3: We compared the learned masks $m$ by visualizing them using volumetric rendering. As shown in the figures above, we observed that the mask learned with uncertainty is cleaner and tends to have a binarized value, which helps avoid the mixture of both tables.

Then the volumetric rendering is applied to each ray at each time step, and a squared error is employed as the loss function:

$$\hat{C}(\boldsymbol{r},t) = \int_n^f T(s,t) \cdot \sigma(\boldsymbol{r}(s),t) \cdot c(\boldsymbol{r}(s),\boldsymbol{d},t)\mathrm{d}s, \quad L_r = \mathop{\mathbb{E}}_{\boldsymbol{r},t}\left[\|C(\boldsymbol{r},t) - \hat{C}(\boldsymbol{r},t)\|_2^2\right]. \quad (4)$$

**Reduce Hash Collisions.** The intuition behind the masked modeling lies in the fact that many parts in the dynamic radiance fields are time-invariant, such as the static objects, the background, etc. These static parts can be well reconstructed from the 3D hash table with a large $m$. For these static points, storing their features in the 4D hash table will occupy a large number of storage, largely increasing the hash collision rate. For these time-invariant parts, the 3D hash encoder can be sufficient to reconstruct, and these static parts will not modify the 4D hash table significantly when $(1 - m(\boldsymbol{x}))$ is small. In this way, the 4D hash table stores the properties of those points which are really dynamic, and those dynamic features in the 4D hash table will be protected by the mask term to suppress the gradients from static points.

**Accelerate Rendering.** Another advantage of using masked space-time hash encoding is to accelerate the fixed view-port rendering. Instead of rendering a novel view video frame-by-frame, we adopt an incremental way of using the mask to avoid redundant computations. Specifically, for a $\mathcal{T}$-frame scene, we first render the initial frame $V_0$ as usual. The obtained mask is used to filter the static parts, and we only render the dynamic parts of other frames with the dynamic weight $(1 - m(x)) > (1 - \epsilon)$, where $\epsilon$ is a hyper-parameter. In this way, except for the initial frame, we only require to render the dynamic part, improving the rendering fps from $1.4$ to $15$, without loss of rendering quality.

The key to the masked space-time hash encoding is that $m(\boldsymbol{x})$ can reflect the dynamics of the point. To this end, we design an uncertainty-guided mask learning strategy to connect the relation between the 3D hash uncertainty $u(\boldsymbol{x})$ and the mask $m(\boldsymbol{x})$. We will elaborate on this strategy in the next part.

### 3.3 Uncertainty-guided Mask Learning

To make the mask $m(\boldsymbol{x})$ well reflect the dynamics of the corresponding point, we design an uncertainty branch in our model to estimate the uncertainty of a point being static, which is a good indicator for the dynamics of the point. To this end, the uncertainty branch is required to reconstruct the dynamic scenes only using the 3D hash table with the input-dependent uncertainties, ignoring the time input. In this way, the dynamic points are regarded as the inherent noise since its supervision are inherently inconsistent (different time step describes different geometries while the uncertainty model is time-agnostic). The inherent noise of dynamic points will lead to a high uncertainty. We adopt the Bayesian learning framework of Kendall and Gal [34] to model the heteroscedastic aleatoric uncertainty for each point.

Specifically, we construct an uncertainty field $u$ with a voxel grid representation. $u(\boldsymbol{x})$ denotes the uncertainty of a space point $\boldsymbol{x}$. We denote the raw output of the uncertainty voxel-grid as $\tilde{u}$, and a soft-plus is used as the activation: $u(\boldsymbol{x}) = u_m + \log\left(1 + \exp(\tilde{u}(\boldsymbol{x}))\right)$, where $u_m$ is a hyper-parameter for shifting the uncertainty values [54]. For each ray, the ray-level uncertainty $\mathrm{U}(\boldsymbol{r})$ is calculated through volumetric rendering [53]:

$$\mathrm{U}(\boldsymbol{r}) = \int_n^f T(s) \cdot \sigma(\boldsymbol{r}(s)) \cdot u(\boldsymbol{r}(s))\mathrm{d}s. \quad (5)$$

Besides, the color and density estimated by the 3D hash table are:

$$\sigma_s(\boldsymbol{x}), \ \boldsymbol{g}_s(\boldsymbol{x}) = \phi_{\theta_1}(\mathrm{h}_{3D}(\boldsymbol{x})), \quad c_s(\boldsymbol{x}, \boldsymbol{d}) = \psi_{\theta_2}(\boldsymbol{g}_s(\boldsymbol{x}), \boldsymbol{d}). \tag{6}$$

The rendered color of this branch is $\hat{C}_s(\boldsymbol{r})$ by applying the volumetric rendering. After that, the uncertainty-based loss for ray $r$ is defined as:

$$L_u = \mathop{\mathbb{E}}_{\boldsymbol{r},t} \left[ \frac{1}{2\mathrm{U}(\boldsymbol{r})^2} \|C(\boldsymbol{r},t) - \hat{C}_s(\boldsymbol{r})\|_2^2 + \log\mathrm{U}(\boldsymbol{r}) \right]. \tag{7}$$

Note that the uncertainty branch is *time-agnostic*, i.e., the predicted color $\hat{C}(\boldsymbol{r})$ is not time-dependent, which is important to make the uncertainty relevant to dynamics. In this way, the uncertainty for each point $u(\boldsymbol{x})$ can be estimated through the above loss function. For the dynamic points, the uncertainty is inherently large because of the ambiguous and inconsistent geometries caused by the inconsistent pixel color supervision. In this perspective, the uncertainty can well reflect the dynamics of each point and could guide the mask values.

**Bridging uncertainty with mask.** Though we find the correlation between the mask $m$ and uncertainty $u$, it is not trivial to connect them. First, $m$ and $u$ are in different value ranges, $m(\boldsymbol{x}) \in [0, 1]$ while $u(\boldsymbol{x}) \in [0, +\infty)$. Second, the distributions of $m$ and $u$ are very different, and the relations between them are non-linear. Imposing a hard relationship between them will impact the training of their specific branches. We instead maximize the mutual information between the two random variables $m$ and $u$ to maximize the nonlinear correlation between them. Although MI can not measure whether the correlation is negative, Eq. (7) can guarantee this. The mutual information $I(m, u)$ describes the decrease of uncertainty in $m$ given $u$: $I(m, u) := H(m) - H(m|u)$, where $H$ is the Shannon entropy. To estimate the mutual information between $m$ and $u$, we adopt the neural estimator in [6] to approximate the mutual information $I(m, u)$ as:

$$I_\Theta(m, u) = \sup_{\theta \in \Theta} \mathbb{E}_{\mathbb{P}_{m,u}} [T_\theta] - \log(\mathbb{E}_{\mathbb{P}_m \otimes \mathbb{P}_u} [e^{T_\theta}]). \tag{8}$$

$\mathbb{P}_{m,u}$ is the joint probability distribution of $m$ and $u$, and $\mathbb{P}_m \otimes \mathbb{P}_u$ is the product of marginals. $T_\theta$ is a neural network with parameters $\theta \in \Theta$. We choose a small MLP with two hidden layers to represent $T_\theta$ and draw samples from the joint distribution and the product marginals to compute the empirical estimation of the expectation. By maximizing the estimated mutual information, we build the correlation between $m$ and $u$. At last, the overall learning objective of MSTH is the combination of the above losses:

$$L = L_r + \lambda \cdot L_u - \gamma \cdot I_\Theta(m, u), \tag{9}$$

where $\lambda$ and $\gamma$ are two hyper-parameters. For rendering a novel view, the uncertainty branches and the mutual information network are disabled.

In practice, the model can learn a reasonable mask $m$ even without the uncertainty guidance. However, this will make the learned mask very noisy and tend to learn a "middle value" in $[0, 1]$, which will make the static points still have a relatively large dynamic weight to modify the 4D table. Fig. 3 visualize the learned mask with or without uncertainty guidance. With uncertainty guidance, the mask will tend to be binarized and the static parts are with very low dynamic weight. For the mutual information constraint, we find it will make the distribution of $m$ towards a Bernoulli distribution which is helpful for reducing hash collision and accelerating rendering speed. Without the constraint, the model tends to learn more from the 3D hash table only even for the dynamic point. This will make the failing captures of some dynamic voxels with transient changes. As a result, the uncertainty guidance will help learn a finer detail, which will be shown in the ablation part.

## 3.4 Ray Sampling

For a natural video, a significant portion of the scene is usually static or exhibits only minor changes in radiance at a specific time across the entire video. Uniformly sampling the space-time rays causes an imbalance between the static and dynamic observations. Therefore, we sample the space-time queries according to the quantification of dynamics. Specifically, we sample a space-time ray $(\boldsymbol{r}, t)$ according to a pre-computed probability $P(\boldsymbol{r}, t)$. We decompose the sampling process into two sub-process: spatial sampling and temporal sampling. Formally, we decompose $P(\boldsymbol{r}, t)$ as $P(\boldsymbol{r}, t) = P(t|\boldsymbol{r})P(\boldsymbol{r})$, which forms a simple Markov process that first sample the space ray according to $P(\boldsymbol{r})$ then sample the time step by $P(t|\boldsymbol{r})$.

Plenoptic Video Dataset  Immersive Dataset  Campus Dataset

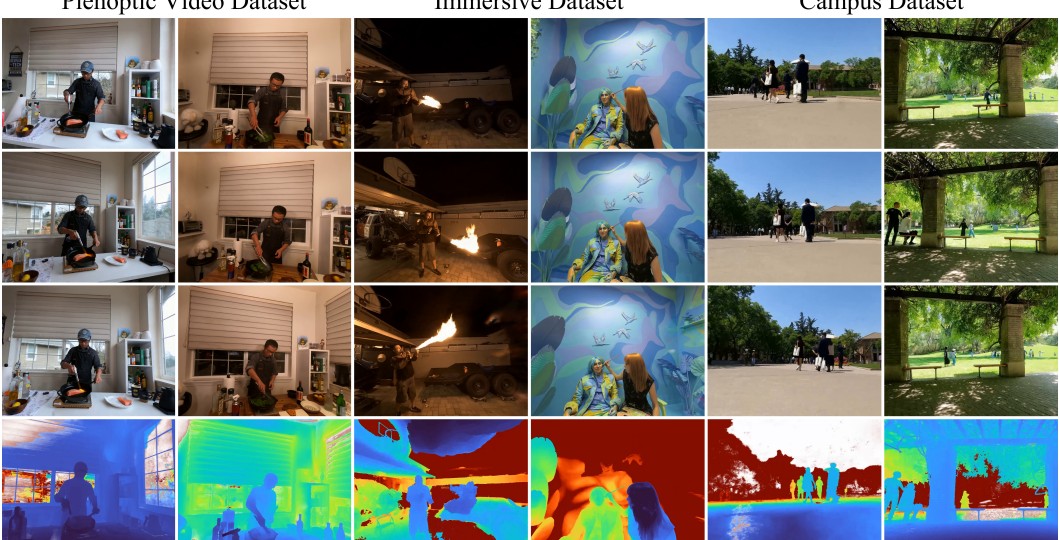

Figure 4: Qualitative results on different datasets of our method. We visualize four novel views in each column, three for RGB and one for depth. For other scenes, we visualize them in the Appendix, and the full spiral videos are presented in our supplemental material.

In practice, we build $P(\boldsymbol{r})$ as the softmax of temporal standard deviation $\texttt{std}(\boldsymbol{r})$ of the corresponding pixel intensities $\{C(r,t)\}_t$:

$$P(\boldsymbol{r}) \propto \exp\left(\texttt{std}(\boldsymbol{r})/\tau_1\right), \tag{10}$$

where $\tau_1$ is the temperature parameter. We set a higher temperature for a softer probability distribution [27]. For temporal sampling, we follow [42] to compute the weight according to the residual difference of its color to the global median value across time $\bar{C}(\boldsymbol{r})$ with temperature $\tau_2$:

$$P(t|\boldsymbol{r}) \propto \exp\left(|C(\boldsymbol{r},t) - \bar{C}(\boldsymbol{r})|/\tau_2\right). \tag{11}$$

### 3.5 Implementation Details

To improve the sampling efficiency, we utilize the proposal sampler [4] that models density at a coarse granularity with a much smaller field, thereby generating more accurate samples that align with the actual density distribution with minor overhead. In our implementation, we use one layer of proposal net to sample 128 points. For the mask, we use a non-hash voxel grid with 128 spatial resolution. To encourage the separation of the static and the dynamic, we utilize a mask loss that aims at generating sparse dynamic voxels by constraining the mask to be close at 1. We also adopt distortion loss [4] with $\lambda_{dist} = 2e - 2$. For uncertainty loss, we set $\gamma = 3e - 4$ and $\lambda = 3e - 5$. We find a small coefficient of mutual information estimator $\gamma$ will have a large impact on the gradients, which is consistent with the observations of [6]. For the hash grid, we implement a CUDA extension based on PyTorch [65] to support the rectangular voxel grid required by the proposed method. For the complete experiment setting and model hyper-parameters, please refer to the Appendix.

## 4 Experiments

### 4.1 Dataset

For validating the performances of the proposed method, we conduct experiments on two public datasets and our collected dataset: **(1)** The Plenoptic Video Dataset [40], which consists of 6 publicly accessible scenes: coffee-martini, flame-salmon, cook-spinach, cut-roasted-beef, flame-steak, and sear-steak. Each scene contains 19 videos with different camera views. The dataset contains some challenging scenes, including objects with topology changes, objects with volumetric effects, various lighting conditions, etc. **(2)** Google Immersive Dataset [9]: The Google Immersive dataset contains light field videos of different indoor and outdoor environments captured by a time-synchronized 46-fisheye camera array. We use the same 7 scenes (*Welder*, *Flames*, *Truck*, *Exhibit*, *Alexa Meade*

Table 1: Quantitative results on Plenoptic Video dataset [40]. We report the average metrics and compare them with other state-of-the-art methods. Our method achieves non-trivial performance improvements on all metrics. * denotes the DyNeRF setting which only reports results on the flame-salmon scene. † denotes the HexPlane setting that removes the coffee-martini scene to calculate average metrics. We report the per-scene metrics in the Appendix.

| Model | PSNR↑ | D-SSIM↓ | LPIPS↓ | Training Time↓ | Rendering FPS ↑ | Storage ↓ |
|---|---|---|---|---|---|---|
| DyNeRF* [40] | 29.58 | 0.020 | 0.099 | 1344 h | < 0.01 | **28MB** |
| Ours* | **29.92** | **0.020** | **0.063** | **20 min** | 15 | 135MB |
| NeRFPlayer [75] | 30.69 | 0.034 | 0.111 | 6 h | 0.05 | - |
| HyperReel [1] | 31.10 | 0.036 | 0.096 | 9 h | 2.0 | 360MB |
| MixVoxels [83] | 30.80 | 0.020 | 0.126 | 80 min | **16.7** | 500MB |
| K-Planes [72] | 31.63 | 0.018 | - | 108 min | - | - |
| Ours | **32.37** | **0.015** | **0.056** | **20 min** | 15 | 135MB |
| HexPlane [10] † | 31.705 | 0.014 | 0.075 | 12 h | - | 200MB |
| Ours† | **33.099** | **0.013** | **0.051** | **20 min** | 15 | 135MB |

*Exihibit*, *Face Paint 1*, *Face Paint 2*) as NeRFPlayer [75] for evaluation. **(3)** To validate the robustness of our method on more complex in-the-wild scenarios, we collect six time-synchronized multi-view videos including more realistic observations such as pedestrians, moving cars, and grasses with people playing. We named the collected dataset as *Campus Dataset*. The Campus dataset is much more difficult than the above two curated ones in the movement complexities and dynamic areas. For detail on the dataset, please see our Appendix. For the above three multi-view datasets, we follow the experiment setting in [40] that employs 18 views for training and 1 view for evaluation. To quantitatively evaluate the rendering quality on novel views, we measure PSNR, DSSIM, and LPIPS [101] on the test views. We follow the setting of [40] to evaluate our model frame by frame. Our method is also applicable to monocular videos. To validate the reconstruction quality with monocular input, we conduct experiments on D-NeRF [68] dataset, which contains eight videos of varying duration, ranging from 50 frames to 200 frames per video. There is only one single training image for each time step. For evaluation, we follow the common setting from [68, 72, 10].

## 4.2 Results

**Multi-view Dynamic Scenes.** We reconstruct the dynamic scenes from multi-view time-synchronized observations on the two public multi-view video datasets and our collected Campus dataset. The quantitative results and comparisons are presented in Tab. 1 and Tab. 2. For the Plenoptic Video dataset, our method surpasses previous state-of-the-art methods by a non-trivial margin, with a $0.7$ to $1.4$ PSNR gains, and $> 30\%$ improvements on the perceptual metric LPIPS. For training, we cost at most $1/5$ training time of other methods, achieving a speedup of $5 - 36$ compared with other fast reconstruction algorithms and a $4000$ speedup compared with DyNeRF [40]. Besides, we also keep a compact model size with only $135$MB of memory occupancy, showing the advantages of the masking strategy which can avoid a large number of hash collisions and make the dynamic hash table size smaller enough. For the Google Immersive dataset, Tab. 2 also shows consistently non-trivial improvements in both training efficiency and reconstruction quality.

Fig. 2 visualizes the qualitative comparisons of our method to other state-of-the-art methods, from which we can observe that our method can capture finer motion details than other methods. For example, MSTH can well reconstruct the fire gun with distinct boundaries and specular reflection, the mark of the hat, and the stripe of the salmon. The Splashing sparks can also be accurately captured. We provide the representative novel-view rendering results of our method in Fig. 4.

**Monocular Dynamic Scenes.** We present the quantitative and qualitative results of our proposed method on the D-NeRF dataset in Tab. 3 and Fig. 5, accompanied by a comparative analysis with related approaches. MSTH outperforms all other methods in terms of SSIM and LPIPS, including methods that focus on dynamic scenes (K-Planes [72] and HexPlane [10]) and TiNeuVox, the current state-of-the-art method for monocular dynamic novel view synthesis. The proposed MSTH achieves superior results without any assumption about the underlying field which demonstrates that our method could be easily extended to accommodate scenarios with varying complexity and dynamics.

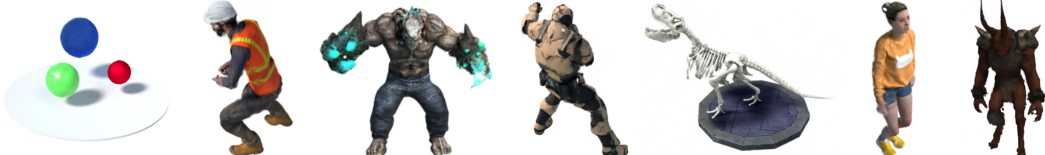

Figure 5: Qualitative results on D-NeRF dataset [68]. The visualized images are rendered from test views. Complete per-scene multi-view images are shown in the Appendix.

Table 2: Quantitive result on Google Immersive dataset [9] and the proposed Campus dataset. Check out the Appendix for detailed comparison and visualization.

| Model | PSNR↑ | SSIM↑ | LPIPS↓ | Training Time ↓ |
|---|---|---|---|---|
| *Google Immersive Video Dataset* | | | | |
| NeRFPlayer [75] | 26.6 | 0.870 | 0.1931 | 6 hrs |
| HyperReel [1] | 28.8 | 0.874 | 0.193 | 2.7 hrs |
| Ours | **29.6** | **0.950** | **0.0929** | **20 min** |
| *Campus Dataset* | | | | |
| Ours | 20.9 | 0.722 | 0.241 | 20 min |

Table 3: Quantitative comparison with related work on D-NeRF dataset [68]. Per-scene metrics are shown in the Appendix.

| Model | PSNR↑ | SSIM↑ | LPIPS↓ |
|---|---|---|---|
| T-NeRF [68] | 29.51 | 0.95 | 0.08 |
| D-NeRF [68] | 30.5 | 0.95 | 0.07 |
| TiNeuVox-B [21] | **32.67** | 0.97 | 0.04 |
| HexPlane [10] | 31.04 | 0.97 | 0.04 |
| K-Planes [72] | 30.84 | 0.96 | - |
| Ours | 31.34 | **0.98** | **0.02** |

### 4.3 Ablation Study

**Ablation on the Masked Hash Encoding.** To evaluate the effect of decomposing a 4D dynamic radiance field into the proposed masked space-time hash encoding, we propose two variants as comparisons: (1) A pure 4D hash encoding. (2) A simple decomposition which is an addition of a 3D hash table and a 4D hash table. Fig. 6 (a) and (b) visualize the qualitative comparisons. From Fig. 6 (a) we can observe that only using the 4D hash table will generate blurry rendering with a 3 PSNR drop. With a simple addition method, the reconstruction quality is improved compared with only a 4D hash table, while the dynamic regions are not captured well. Fig. 6 (b) and (c) show the comparisons of MSTH with the addition. MSTH improves the LPIPS by $20\%$ compared with the addition method.

**Ablation on the uncertainty-based guidance.** We visualize the learned mask with or without uncertainty-based guidance, Fig. 3 shows the comparison. The learned mask with uncertainty loss will distinguish the static and dynamic parts more clearly, with less noisy judgment, leading to an assertive distribution on the mask, which is beneficial to avoid hash collisions. Fig. 6 (d) visualize the qualitative comparisons. MSTH with uncertainty loss performs better on the motion details.

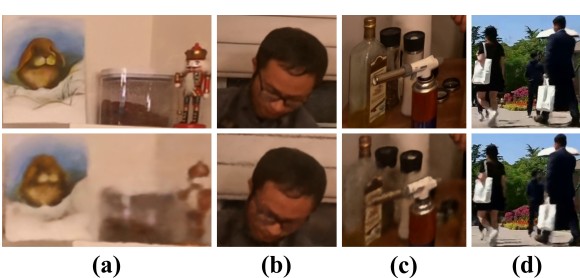

|  (a)  |  (b)  |  (c)  |  (d)  |

Figure 6: Qualitative comparisons for different ablations. Zoom in for a better inspection.

## 5 Limitations and Conclusion

**Limitations.** MSTH tends to generate unsatisfying results when detailed dynamic information is insufficient, especially in monocular dynamic scenes, since it relies solely on the input images. When the input is blurred, occluded, or incomplete, the method may struggle to infer the motion information, leading to artifacts such as ghosting, flickering, or misalignment, which may degrade the visual quality. Further research is needed to develop advanced methods that handle complex scenes, motion dynamics and integrate multiple information sources to enhance synthesis accuracy and robustness.

**Conclusion.** In this paper, we propose a new method to reconstruct dynamic 3D scenes in a time- and space-efficient way. We decouple the representations of the dynamic radiance field into a time-invariant 3D hash encoding and a time-variant 4D hash encoding. With an uncertainty-guided mask as weight, our algorithm can avoid a large number of hash collisions brought about by the additional time dimension. We conduct extensive experiments to validate our method and achieve state-of-the-art performances with only 20 minutes of training. Besides, we collect a complex in-the-wild multi-view video dataset for evaluating the robustness of our approach on more realistic dynamic

scenes, including many daily activities. We hope our method can serve as a fast and lightweight baseline to reconstruct dynamic 3D scenes, which may derive many valuable applications such as interactively free-viewpoint control for movies, cinematic effects, novel view replays for sporting events, and other VR/AR applications.

# 6 Acknowledgement

This work was supported in part by the National Natural Science Fund for Distinguished Young Scholars under Grant 62025304. Thank Xueping Shi for giving helpful discussion.

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
