# OpenReview forum: "Masked Space-Time Hash Encoding for Efficient Dynamic Scene Reconstruction"
_NeurIPS.cc/2023/Conference — NeurIPS 2023 spotlight_

### Official Review · Reviewer_ohyg · 2023-07-02

**Soundness:** 3 good
**Presentation:** 4 excellent
**Contribution:** 3 good
**Rating:** 7
**Confidence:** 5

**Summary:**

This paper proposes the masked space-time hash encoding to efficiently reconstruct dynamic scenes. The insights behind the paper is that: most part of the scene is static, simply modelling such static parts can dramatically increase the probability of hash collisions; while increasing the hash table entries requires more memory. To solve these issues, this paper proposes to decouple the scene into a static part and a dynamic part, where the two parts are jointly learned through an uncertainty mask. The uncertainty mask is modeled by an individual voxel grid, on which each voxel stores an value of the uncertainty field. To bridge the gap between uncertainties and masks, a neural estimator is adopted to approximate the mutual information. Experiments are conducted on the Plenoptic Video Dataset, the Google Immersive Dataset, and the self-collected time-synchronized multi-view videos. As shown in the quantitative results, MSTH surpasses state-of-the-art dynamic NeRF methods (HexPlane, K-Planes, etc) in terms of both the reconstruction quality and training/rendering time. Ablation studies also shows the effectiveness of the proposed masked space-time hash encoding.

**Strengths:**

This is a technically very solid paper. Compared to previous methods, the proposed method can reconstruct dynamic scenes with higher quality while highly reduced the training time. I like the idea of using uncertainty to mask the static part and dynamic part in a scene instead of simply a combination of 3D voxel grid and a 4D voxel grid. The experiments are exhaustive and highly support the effectiveness of the proposed method.

**Weaknesses:**

(1) In Figure 3 (c), it seems some static parts have high uncertainty values (the top-left part and the middle-right part)?


**Questions:**

- (1)  Is Figure 3(c) revealing that the uncertainty does not accurately model the static part and dynamic part?
- (2) In Section 4.3, the authors proposed two variants for ablation study: (1) A pure 4D hash encoding. (2) A simple decomposition which is an addition of a 3D hash table and a 4D hash table. Actually, for the second variant, I'm not quite sure whether the authors split the scene into a static part or a dynamic part using additional models (for example, using the **segment anything model (SAM)** to split a non-rigid object/human from the static background), and then separately model them. If it is not, I wonder would it better to separately model the dynamic part and static part with some pre-computed masks instead of jointly learn the uncertainty during training?

**Limitations:**

The authors address the major limitations of existing dynamic NeRF methods, e.g. the efficiency and rendering quality. Some other limitations, such as reconstructing monocular dynamic scenes, blurry scenes, etc, are mentioned in the paper.

---

> ### Author Rebuttal · Authors · 2023-08-08
>
> > Q1: Is Figure 3 \(c\) revealing that the uncertainty does not accurately model the static part and dynamic part?
>
>
> The dynamic region inferred by the model may include some noise induced by algorithm-irrelevant reasons, such as inaccurate estimated camera poses and parameters, lack of key points when the scene contains little high-frequency information and etc. In our experiment, the upper-left part of the scene could not be well reconstructed even by exploiting per-frame static nerf algorithms like Instant-NGP. As a result, we consider this as a data intrinsic noise that has little impact on the visual quality.
>
> > Q2: In Section 4.3, the authors proposed two variants for ablation study: (1) A pure 4D hash encoding. (2) A simple decomposition which is an addition of a 3D hash table and a 4D hash table. Actually, for the second variant, I'm not quite sure whether the authors split the scene into a static part or a dynamic part using additional models (for example, using the segment anything model (SAM) to split a non-rigid object/human from the static background), and then separately model them. If it is not, I wonder would it better to separately model the dynamic part and static part with some pre-computed masks instead of jointly learn the uncertainty during training?
>
> For the variant "A simple decomposition which is an addition of a 3D hash table and a 4D hash table" in the ablation study part, we did not apply any off-the-shelf segmentation model to split the dynamic part from the static part. In this ablation study, the separation of the static and dynamic part is learned implicitly through reconstruction loss without the guidance of uncertainty, which results in sub-optimal performance as shown in Fig 6 in the paper, and proves the effectiveness of the proposed uncertainty estimation.
>
> Incorporating segmentation models in the separation of static and dynamic regions introduces a strong prior that the dynamic region consists of semantically-grounded objects. Besides, the identification of dynamic objects requires manual specification. This assumption is not reasonable in many situations. For example, in the *flame salmon* scene in the Plenoptic dataset, some foreground objects (including the cups, the table and etc.) are actually static. Apart from that, some dynamic regions may not manifest within the vocabulary of a particular segmentation algorithm. Even if the dynamic part can be roughly captured, the concrete bounds cannot be well inferred without other guidance (e.g. the hands and head of the cooking man in the *flame salmon* scene), and tracking the dynamic objects in multiple videos also introduces extra complexity.
>
> In contrast, our solution provides a more general framework that requires no prior on the dynamic scenes, in which the granularity of dynamic region is automatically determined under the guidance of uncertainty.

---

> > ### Comment · Area_Chair_UaHs · 2023-08-18
> >
> > Dear Reviewer,
> >
> > Since the discussion with authors is closing soon, could you please go over the reviews and rebuttals, and respond to the content of the authors response with a message to the authors (you can post with one message summarizing all such reviews). It is important that authors receive a reply to their rebuttals, as they have tried to address comments raised by the reviewers.
> >
> > -AC

---

> > ### Comment · Reviewer_ohyg · 2023-08-20
> > **Thanks for the rebuttal**
> >
> > Thanks to the authors for the rebuttal. All of my concerns are addressed. Then I decided to keep my current rating for this paper.

---

### Official Review · Reviewer_Vvsv · 2023-07-04

**Soundness:** 4 excellent
**Presentation:** 4 excellent
**Contribution:** 3 good
**Rating:** 7
**Confidence:** 4

**Summary:**

The paper presents MSTH, a method that efficiently reconstructs dynamic 3D scenes from multi-view or monocular videos. The proposed solution uses a space-time hash encoding, a prediction of masks, and uncertainty values that help a method to identify 3D points that belong to moving objects. The intuition is that this masking and uncertainty will help a method identify static points which are easy to reconstruct but also handle well the moving points. Lastly, the formulation of the method allows it to render in fast manner also preserving the size of the model. The paper presents experiments on three real-datasets: Plenoptic Video Dataset, Google Immersive Dataset, and a newly introduced dataset. The presented experiments show that the MSTH improves the PSNR while keeping both the training time and inference time short as well as maintaining a constant size model.

**Strengths:**

In sum, I find the paper to be well executed and explained. Overall, I think the problem of reconstructing dynamic scenes is a challenging but important problem to solve. In detail here are the strengths I find:

S1. I find the use of masking to identify the static and dynamic points an interesting and simple idea. I think the idea has good intuitive rationale and it should be effective.

S2. The inclusion of uncertainty in the formulation I think is an important example of why uncertainty is very important in machine learning. I support the idea of combining uncertainty with the mask prediction as a mean to improve results.

S3. The experiments use real-data and challenging scenes to showcase the benefits of the proposed approach. Overall, I think the experiments are well executed and highlight the benefits of the MSTH.

S4. The clarity and description of MSTH is good. Overall, I think the proposed method should be easy to reproduced given the level of clarity of the narrative.

**Weaknesses:**

Overall, I find the paper to be well executed and clear. I only have one weakness to point out:

W1. I think the paper is lacking a more in-depth discussion about the newly created dataset to test MSTH. Whil I understand that the algorithm is important, I also think the data part is as important as the algorithm. At the end of the day, the data is required and crucial to solve many problems in the era of deep learning. Thus, I suggest that a final version of the paper includes an extended discussion of the dataset curated by the authors.

While I like the approach and explanation of MSTH, I do have some minor concerns:

W2. The uncertainty described in the method in a way breaks with the intuitive understanding that an uncertainty value corresponds to a confidence value ranging from 0-1 and could be interpreted as a probability of correctness. However, the "uncertainty" introduced in Section 3.3 deviates from this common understanding. I think the paper should explicitly state a concrete meaning of uncertainty to clarify that the common understanding of uncertainty does not apply to this narrative. The main reason I bring this up is because I think this can confuse readers in the future.

W3. In several parts of the narrative (e.g., lines 188 and 186) the paper uses the term "dynamics of a point". I find this term to be confusing since in many fields (e.g., robotics or control) the dynamics typically refer to a model that describes how the state of something changes over time. I think the paper should clarify what this exactly means to clear any misconception of this term.

W4. Missing definition of different terms: 1) $\tilde{m}(\cdot)$ is never defined in the text; 2) what does "large $m$" means in line 168? Isn't $m$ the learned mask? Or is it a value?

----
Post Rebuttal

After the rebuttal, all my concerns were clarified and I still think this is a good contribution. Thus, I maintain my rating.

**Questions:**

Please see Weaknesses section.

**Limitations:**

I think the paper clearly states the limitations of the approach.

---

> ### Author Rebuttal · Authors · 2023-08-08
>
> > W1: I think the paper is lacking a more in-depth discussion about the newly created dataset to test MSTH. While I understand that the algorithm is important, I also think the data part is as important as the algorithm. At the end of the day, the data is required and crucial to solve many problems in the era of deep learning. Thus, I suggest that a final version of the paper includes an extended discussion of the dataset curated by the authors.
>
>
> Due to the line limit for the submission, we only provide a concise description of the proposed Campus dataset in the main text. A more detailed introduction and configuration can be found in the Appendix. We would provide a more comprehensive explication of the dataset within the main body.
>
>
> > W2: The uncertainty described in the method in a way breaks with the intuitive understanding that an uncertainty value corresponds to a confidence value ranging from 0-1 and could be interpreted as a probability of correctness. However, the "uncertainty" introduced in Section 3.3 deviates from this common understanding. I think the paper should explicitly state a concrete meaning of uncertainty to clarify that the common understanding of uncertainty does not apply to this narrative. The main reason I bring this up is because I think this can confuse readers in the future.
>
> The uncertainty we used in this paper is derived from the *Aleatoric Uncertainty* proposed in [1] which model the intrinsic noise of the observed data by Gaussian distributions. The uncertainty $U(\cdot)$ stands for the learned standard deviation of the underlying Gaussian which is unbounded by its definition. The uncertainty augmented loss would assign a relatively large uncertainty value on those spatial points with high time variance which helps the model in the separation of static and dynamic regions.
>
> [1] Kendall, Alex, and Yarin Gal. "What uncertainties do we need in Bayesian deep learning for computer vision?." Advances in neural information processing systems 30 (2017).
>
>
> > W3: In several parts of the narrative (e.g., lines 188 and 186) the paper uses the term "dynamics of a point". I find this term to be confusing since in many fields (e.g., robotics or control) the dynamics typically refer to a model that describes how the state of something changes over time. I think the paper should clarify what this exactly means to clear any misconception of this term.
>
>
> We thank the reviewer for pointing out the confusion arising from the lack of clarity in our submission. In the context of our paper, "dynamics of a point"
> to how frequently the state of density and color changes through time, e.g, a more dynamic point will have more density and color states through time, which can be reflected by the uncertainty. In contrast, the traditional understanding of dynamics within robotics and control refers to a model explicitly formulated and constructed by human intervention, which is devised to capture and predict the behavior of an object. In response to the concerns raised regarding potentially misleading representations, we assert our commitment to rectify ambiguities and refine the clarity of terminology in the refined version of our paper.
>
> > W4: Missing definition of different terms: 1) $\tilde{m}(\cdot)$ is never defined in the text; 2) what does $m$ means in line 168? Isn't the learned mask? Or is it a value?
>
> We thank the reviewer for pointing out the potential missing definitions. The symbol $\tilde{m}$ in Eq(2) refers to the unactivated mask value, namely the raw output of the mask branch of the neural network. This $\tilde{m}(\cdot)$ is converted to the final mask value by $\text{sigmoid}$ function, as shown in Eq(2).
> The $m$ in line 168 stands for the same meaning as that in other equations, namely the corresponding mask value. Our symbol system is consistent in the paper, the $m$ is firstly defined in Eq(2) and retains the same meaning in line 168.

---

> > ### Comment · Reviewer_Vvsv · 2023-08-12
> > **RE: Rebuttal by Authors**
> >
> > Thanks for clarifying. Please make sure these clarifications are discussed and included in a final manuscript.

---

### Official Review · Reviewer_hvP1 · 2023-07-05

**Soundness:** 3 good
**Presentation:** 3 good
**Contribution:** 3 good
**Rating:** 6
**Confidence:** 4

**Summary:**

This paper present a method for efficient dynamic scene reconstruction. They represent the dynamic scene with a weighted combination of a 3D hash encoding (for static part) and a 4D hash encoding (for dynamic region). The weight is learnable and can be represented by a multi-resolution hash table or a 3D voxel grid. For each query 3D point, they can then get a weighted feature encoding from the hash tables. The photometric loss is then used to supervise the training.

To better supervise the learnable mask/weight representation, they further propose an uncertainty guided mask learning and exploit the mutual correlation between "3D point uncertainty" and "3D point mask" to supervise the learning of the mask representation.

The experimental evaluations are conducted on two public datasets and a self-collected dataset. The experimental results demonstrate the effectiveness of the proposed method.

**Strengths:**

1: The paper presents a novel method to improve both the reconstruction quality and efficiency for dynamic scenes. The experimentral results demonstrate its effectiveness.

2: The paper is well written.

**Weaknesses:**

I do not find severe flaws of the paper and would like to discuss with other reviewers if necessary.

**Questions:**

The mask representation is not time dependent and the uncertainty supervision is based on the inconsistency between the rendered images & the real captured images. I have following two questions:

1: Since the 3D scene is dynamic, the mask(static/dynamic) for each 3D position should also depend on time. Why you choose to use a time-independent representation.

2: Is it like that as long as the 3D point being occupied by a dynamic object (even for partial frames), it will also be predicted as dynamic (which would have a high uncertainty from Eq. 7)? If so, will your method be able to work well if the dynamic object occupy a larger portion of the image and sweep over most of the 3D space across time. To what extent it can perform well?

**Limitations:**

If Question 2 is right, please address the corresponding limitation in Section 5 of the main paper.

---

> ### Author Rebuttal · Authors · 2023-08-08
>
> > Q1: Since the 3D scene is dynamic, the mask(static/dynamic) for each 3D position should also depend on time. Why you choose to use a time-independent representation.
>
> We design the time-independent mask mainly due to the inference efficiency. Specifically, we find the efficiency bottleneck of the hash encoder is mainly from the hash mapping (instead of the mlp due to the fast tiny-cuda-nn implementation). And for the hash mapping, the bottleneck is the memory access time. For a 3D mask, the hash mapping requires to access eight times of memory for a trilinear interpolation. If we use a 4D time-dependent mask, the hash mapping requires to access 16 times of memory for a 4-linear interpolation, which makes the inference time much slower than 3D masks. Besides, due to the extra time dimension, a 4D mask is also memory-inefficient while the performances we evaluated are similar with a 3D mask.
>
> Concretely, we also conduct experiments to compare the time-dependent and time-independent versions of masks. We compare them in the following table (on the Plenoptic Video dataset).
>
> | Metrics | PSNR | LPIPS | FPS | Memory |
> |:-------:|:----:|:-----:|:---:|:------:|
> | 3D Mask | 33.1 | 0.051 | 15  |  135M  |
> | 4D Mask | 32.7 | 0.053 |  9  |  183M  |
>
>
> > Q2: Is it like that as long as the 3D point being occupied by a dynamic object (even for partial frames), it will also be predicted as dynamic (which would have a high uncertainty from Eq. 7)? If so, will your method be able to work well if the dynamic object occupy a larger portion of the image and sweep over most of the 3D space across time. To what extent it can perform well?
>
> In our implementation, if the dynamics of the underlying scene are of great complexity, our solution is to allocate more space for dynamic part by using a larger hash table. The main problem of the 4D mask is the inference efficiency which is mentioned in Q1.
>
> We collect some complex multi-view videos in our proposed Campus dataset which exhibits long and large dynamic areas in our supplemental video. The proposed MSTH demonstrates commendable performance in the aforementioned scenes, despite the utilization of a limited-size dynamic hash table. This observation substantiates the algorithm's efficacy in handling scenes that encompass significant dynamic areas.
>
> For most routine scenes, the 3D mask is enough for effectiveness and efficiency. We also believe the 4D time-dependent mask will be more memory efficient for extremely complex scenes, e.g., the dynamic areas are very large. We believe the more fine-grained 4D masks are a better solution in this situation for memory efficiency. We will also explore this situation more and sincerely thank the reviewer for giving this valuable feedback.

---

> > ### Comment · Area_Chair_UaHs · 2023-08-18
> >
> > Dear Reviewer,
> >
> > Since the discussion with authors is closing soon, could you please go over the reviews and rebuttals, and respond to the content of the authors response with a message to the authors (you can post with one message summarizing all such reviews). It is important that authors receive a reply to their rebuttals, as they have tried to address comments raised by the reviewers.
> >
> > -AC

---

> > ### Comment · Reviewer_hvP1 · 2023-08-20
> >
> > Thanks for the clarifications. It is much clearer now. I would keep my original rating.

---

### Official Review · Reviewer_r2SF · 2023-07-05

**Soundness:** 3 good
**Presentation:** 2 fair
**Contribution:** 2 fair
**Rating:** 5
**Confidence:** 4

**Summary:**

This paper tries to address the challenge of efficiently representing 3D dynamic scenes. It proposes a decoupled representation that uses separate neural implicit representations for dynamic and static 3D points. This approach can reduce hash collision and save the storage of the multi-level hash feature grids.

**Strengths:**

1. An automatic learning procedure to detect dynamic points in the 3D scene. This is achieved by integrating Bayesian learning to treat dynamic points as noise and reduce their weights in the photo-consistency loss.
2. A separate 3D and 4D hash feature grids to save the storage.
3. The optimization of mutual information between uncertainty and mask is demonstrated to be beneficial to the rendering quality.

**Weaknesses:**

1. The concept of representing scenes with separate static and dynamic parts has been investigated in NeRFPlayer or NeRF in the wild.  Integrating Bayesian learning to predict the dynamic points is also not new.
2. Lack quantitative ablation study on the mutual information optimization, which weaken the contribution of this step.
3. Inadequate references. There are missing references related to neural implicit representation for large scale and dynamic scenes:
 Xiuchao Wu et al., Scalable Neural Indoor Scene Rendering, SIGGRAPH 2022
 Haithem Turki et al., Mega-NERF: Scalable Construction of Large-Scale NeRFs for Virtual Fly-Throughs, CVPR 2022
 Liao Wagn et al., Fourier PlenOctrees for Dynamic Radiance Field Rendering in Real-time, CVPR 2022 Oral

**Questions:**

In Fig.6, what is the meaning of each row?

**Limitations:**

The proposed method is most effective for scenes with a significant amount of static parts. For scenes with a small amount of static parts, the storage savings will be less significant.

---

> ### Author Rebuttal · Authors · 2023-08-08
>
> Weaknesses:
> 1. >1.1.The concept of representing scenes with separate static and dynamic parts has been investigated in NeRFPlayer or NeRF in the wild.
>
>     Although the abstract concept of separating static and dynamic is not novel, many existing methods like NeRFPlayer, NeRFW, and MixVoxels adopt this strategy. We emphasize that our separation differs from previous approaches both in purpose (compact storage) and means (estimated by uncertainty). Our separation is soft which could help use very small hash tables to represent large numbers of spatiotemporal voxels without disastrous hash collisions, which improve the memory and rendering efficiency. In contrast, NeRFPlayer used three branches to ensemble the results for improving quality, while their design can not improve either memory or rendering efficiency. We summarize the differences in the following Table.
>
> |                 |    NeRFPlayer     |    NeRF-W     |           MSTH(Ours)           |
> |:---------------:|:-----------------:|:-------------:|:------------------------------:|
> |     Purpose     |  improve quality  | remove noises | compactness (small hash table) |
> | How to separate | learnable network |  uncertainty  | uncertainty + MI maximization  |
>
>
> > 1.2 Integrating Bayesian learning to predict the dynamic points is also not new.
>
> We indeed take inspiration from the successful work of NeRF-W. However,
> NeRF-W and our method aim to solve different problems, NeRF-W use uncertainty to reduce the impact of transient objects (noise), while they do not consider the reconstruction of transient objects. Instead, we treat the dynamic point as noise and they are also required to be reconstructed. With this important difference, we are required to make the uncertainty represent the dynamics through mutual information maximization.
>
> > 2. Lack quantitative ablation study on the mutual information optimization, which weaken the contribution of this step.
>
> In Table 1 in the appendix, we ablate the Mutual information optimization by substituting the mutual information term with other descriptors of correlation including the Pearson correlation coefficient and a hard-coded linear correlation. Results demonstrate the effectiveness of the proposed Mutual information correlation. For clarity, we paste the comparison in the following table.
>
>
>
> | Correlation | PSNR$\uparrow$ | LPIPS$\downarrow$ |
> |:-----------:|:--------------:|:-----------------:|
> |    Mask     |     29.64      |       0.087       |
> |   Pearson   |     29.87      |       0.107       |
> | Hard-coded  |     28.74      |       0.124       |
> |   **MI**    |   **29.93**    |     **0.063**     |
>
>
> > 3. Inadequate references. There are missing references related to neural implicit representation for large scale and dynamic scenes: Xiuchao Wu et al., Scalable Neural Indoor Scene Rendering, SIGGRAPH 2022 Haithem Turki et al., Mega-NERF: Scalable Construction of Large-Scale NeRFs for Virtual Fly-Throughs, CVPR 2022 Liao Wagn et al., Fourier PlenOctrees for Dynamic Radiance Field Rendering in Real-time, CVPR 2022 Oral
>
> For the missing references, they are indeed related and we will include them in the related work. (Fourier PlenOctrees [85] is already included in our draft)

---

> > ### Comment · Area_Chair_UaHs · 2023-08-18
> >
> > Dear Reviewer,
> >
> > Since the discussion with authors is closing soon, could you please go over the reviews and rebuttals, and respond to the content of the authors response with a message to the authors (you can post with one message summarizing all such reviews). It is important that authors receive a reply to their rebuttals, as they have tried to address comments raised by the reviewers.
> >
> > -AC

---

> > > ### Comment · Reviewer_r2SF · 2023-08-22
> > >
> > > Thanks for the clarification.  I will update my rating to borderline accept.  Please incorporate the materials in the rebuttal to explain the contribution of this paper in a more clear way.

---

### Decision · Program_Chairs · 2023-09-21

**Decision:**

Accept (spotlight)

**Comment:**

The final score of this paper is A, A, WA, BA. All reviewers agree that this paper has made contributions and the proposed method is effective in speeding up the training and reducing the memory storage required for dynamic scene reconstruction. The rebuttal has done an excellent job in providing additional experiments and explanations. Please incorporate all the additional materials in rebuttal into the final version of the paper.